# Actin-Related Protein 4 and Linker Histone Sustain Yeast Replicative Ageing

**DOI:** 10.3390/cells11172754

**Published:** 2022-09-03

**Authors:** Mateusz Mołoń, Karolina Stępień, Patrycja Kielar, Bela Vasileva, Bonka Lozanska, Dessislava Staneva, Penyo Ivanov, Monika Kula-Maximenko, Eliza Molestak, Marek Tchórzewski, George Miloshev, Milena Georgieva

**Affiliations:** 1Department of Biochemistry and Cell Biology, Institute of Biology and Biotechnology, University of Rzeszow, 35-601 Rzeszow, Poland; 2Laboratory of Yeast Molecular Genetics, Institute of Molecular Biology “Acad. R. Tsanev”, Bulgarian Academy of Sciences, 1123 Sofia, Bulgaria; 3The Franciszek Górski Institute of Plant Physiology, Polish Academy of Sciences, 30-239 Kraków, Poland; 4Department of Molecular Biology, Maria Curie-Skłodowska University, 20-033 Lublin, Poland

**Keywords:** ageing, actin-related protein 4, Arp4p, linker histone, Hho1p, replicative lifespan

## Abstract

Ageing is accompanied by dramatic changes in chromatin structure organization and genome function. Two essential components of chromatin, the linker histone Hho1p and actin-related protein 4 (Arp4p), have been shown to physically interact in *Saccharomyces cerevisiae* cells, thus maintaining chromatin dynamics and function, as well as genome stability and cellular morphology. Disrupting this interaction has been proven to influence the stability of the yeast genome and the way cells respond to stress during chronological ageing. It has also been proven that the abrogated interaction between these two chromatin proteins elicited premature ageing phenotypes. Alterations in chromatin compaction have also been associated with replicative ageing, though the main players are not well recognized. Based on this knowledge, here, we examine how the interaction between Hho1p and Arp4p impacts the ageing of mitotically active yeast cells. For this purpose, two sets of strains were used—haploids (WT(n), *arp4*, *hho1Δ* and *arp4 hho1Δ*) and their heterozygous diploid counterparts (WT(2n), *ARP4*/*arp4*, *HHO1*/*hho1Δ* and *ARP4 HHO1*/*arp4 hho1Δ*)—for the performance of extensive morphological and physiological analyses during replicative ageing. These analyses included a comparative examination of the yeast cells’ chromatin structure, proliferative and reproductive potential, and resilience to stress, as well as polysome profiles and chemical composition. The results demonstrated that the haploid chromatin mutants *arp4* and *arp4 hho1Δ* demonstrated a significant reduction in replicative and total lifespan. These findings lead to the conclusion that the importance of a healthy interaction between Arp4p and Hho1p in replicative ageing is significant. This is proof of the concomitant importance of Hho1p and Arp4p in chronological and replicative ageing.

## 1. Introduction

Ageing is a complex process comprising an inevitable deterioration in cell functionality, and is considered a universal, intrinsic, progressive and deleterious process [1,2]. Dozens of theories have emerged to describe its cause and molecular basis, such as the free radical theory of ageing (TOA), mitochondrial, information-dependent, evolutionary, inflammation-dependent, waste accumulation TOA, etc. [1,2,3,4,5]. Hitherto, no single theory can explain the complexity of the changes observed during ageing at cellular and organismal levels. Nine hallmarks of ageing have been postulated to characterize the process, including four primaries (genome unsteadiness, telomere shortening, epigenetic changes and loss of proteostasis), three antagonistic (mitochondrial dysfunction, deregulated nutrient sensing and cellular senescence) and two integrative (altered intercellular communication and stem cell exhaustion) hallmarks [6,7]. López-Otín and co-authors have described links between each of the nine hallmarks of ageing and the undesirable metabolic alterations during this process [7], and only recently it has been stated that “a subtle change in metabolism has an obvious effect on the rate of replicative senescence” [1]. Moreover, these changes in the metabolism during ageing were linked with changes in chromatin organization and dynamics [8,9].

Budding yeast is an excellent research model in many fields of biology and medicine, including ageing. As a unicellular organism, it has many useful features that allow it to be used successfully in the genetic aspect of ageing research. Additionally, easy genetic manipulations and fast growth make yeast a workhorse in longevity studies [10,11]. Yeast cells live in two ploidy states, haploids or diploids, and reproduce asexually and asymmetrically by budding. This biological aspect is important for analysing changes in gene expression in haploid and diploid heterozygous systems. A unicellular organism, *S. cerevisiae*, has also been useful in the detection of preserved ageing pathways, such as dietary restriction, oxidative stress response, and TOR signalling [12,13]. Moreover, yeast is an invaluable model organism that provides the opportunity to clearly distinguish between replicative and chronological ageing. The chronological lifespan (CLS) defines the ageing of non-budding, postmitotic cells. It typically begins about the third day after the culture has been started and is evaluated by the time these cells remain viable in the stationary phase [14]. The replicative lifespan (RLS) is characterized by the ageing of mitotically active cells and is determined by the number of daughter cells (buds) produced by a mother [11]. In *S. cerevisiae*, the duration of RLS depends on the strain and averages about 24 doublings. It is important to emphasize that replicative ageing is a deficiency in the replicative ability of mother cells when growing on a rich medium, while chronological ageing occurs under food shortages during the stationary phase and reflects the viability of postmitotic cells [3]. Therefore, the two types of ageing differ not only temporally but also biochemically. Both of these ageing models have played a role in discovering factors promoting ageing, e.g., mitochondrial dysfunction, which are evolutionarily conserved [15]. They are key to understanding the ageing of higher organisms [3,10,16].

Chromatin structure changes in ageing [17,18]. Recent data have accumulated and proved that alterations in genome organization can also lead to premature ageing phenotypes in mitotically inactive cells [19,20,21,22]. Two essential components of chromatin, the linker histone Hho1p and the actin-related protein 4 (Arp4p), have been shown to physically interact in *S. cerevisiae* cells, thus maintaining chromatin dynamics and function, as well as genome stability and cellular morphology during chronological yeast ageing [20,23]. Disrupting this interaction has been demonstrated to influence the stability of the yeast genome and its activity, thus reducing the ability of cells to withstand stress [19]. It has been further proven that the abrogated interaction between these two chromatin proteins elicits premature ageing phenotypes in chronologically ageing yeast cells, thus highlighting the importance of the proper chromatin structure for normal gene expression, cell homeostasis and physiology.

Based on this knowledge, we decided to examine how the interrupted interaction between Hho1p and Arp4p would impact the ageing of mitotically active yeast cells. For this purpose, two sets of strains were used: haploids (WT(n), *arp4*, *hho1Δ* and *arp4 hho1Δ*) and their heterozygous diploid counterparts (WT(2n), *ARP4/arp4*, *HHO1/hho1Δ* and *ARP4 HHO1/arp4 hho1Δ*). In heterodiploids, one allele of the studied genes of *HHO1* (coding for the yeast linker histone, Hho1p) and *ARP4* was mutated, while the other was a wild type. We performed extensive morphological and physiological analysis during replicative ageing. These analyses included a comparative examination of the yeast cells’ chromatin structure, proliferative and reproductive potential, and resilience to stress, as well as an examination of polysome profile and the yeast cells’ chemical composition. Our results demonstrate that the haploid chromatin mutants *arp4* and *arp4 hho1Δ* result in a significant reduction in replicative and total lifespan. These findings highlight our conclusion that the importance of a healthy interaction between Arp4p and Hho1p in replicative ageing is significant and that in the haploid mutant background this is decisively pronounced. The latter is proof of the concomitant importance of Hho1p and Arp4p in ageing.

## 2. Materials and Methods

### 2.1. Strains and Growth Conditions

The studied *S. cerevisiae* yeast strains are listed in Table 1.

The cells were cultivated in a regular liquid-rich YPD medium (1% Difco™ Yeast Extract, 1% Yeast Bacto-Peptone, and 2% glucose; Becton, Dickinson U.K. Limited). The yeast cultures were grown on a rotary shaker at 150 rpm or on a solid YPD medium with 2% agar. Unless otherwise specified, cultivation was carried out at 28 °C.

It should be noted that the standard name of the gene encoding the actin-related protein 4 is *ARP4* with the systematic name *YJL081C* and alias *ACT3* (https://www.yeastgenome.org/locus/S000003617; accessed on 25 August 2022). The gene coding for the histone one protein has the standard name *HHO1* and the systematic name *YPL127C* (https://www.yeastgenome.org/locus/S000006048; accessed on 25 August 2022).

### 2.2. Yeast Mating Experiments

The wild-type haploid *MAT*a and the isogenic mutants *arp4*, *hho1Δ*, *arp4 hho1Δ* and the wild-type *MAT*α BY4742 strain were grown in a rich YPD liquid medium overnight under aeration. Then, 200 μL of *MAT*α cell suspension was added to 200 μL of each *MAT*a strain in a 1.5 mL microcentrifuge tube. The tubes were vortexed to fully mix the cells and incubated at 30 °C overnight (16–20 h) without shaking. Next, 50 μL aliquots of the mating yeast cultures were spread on the appropriate solid Synthetic Complete medium (SC): 2% dextrose; 1.7% yeast nitrogen base and supplements according to the auxotrophic requirements of the strains. for the selection of diploids (Becton, Dickinson U.K. Limited). Plates were hatched at 30 °C for 3 to 5 days to permit diploid cells to form colonies. The yeast diploids were confirmed by their auxotrophic phenotypes.

### 2.3. Assessment of the Yeast Cell Growth Rate

A liquid YPD medium was used for the growth assays. Cell suspensions of yeast were incubated at 28 °C for 12 h in a Heidolph incubator 1000 (Heidolph Instruments GmbH & Co. KG, Schwabach, Germany) at 1200 rpm. The cell growth was observed by the Anthos 2010 type 17550 microplate reader (Biochrom Ltd., Cambridge, UK) at 600 nm (OD_600_). Measurements were performed at 2 h intervals for a period of 12 h.

Additionally, the number of cells/mL in each culture was calculated by a Malassez chamber (Fisher Scientific, Waltham, MA, USA). The one-way ANOVA was applied for statistical data evaluations and *p*-values of 0.001 were considered significant.

### 2.4. Calculation of the Mean Doubling Time

The mean doubling time was analysed for each cell as described before [26]. During the routine determination of the yeast budding lifespan, the mean doubling time of the analysed cells was determined. The presented data are the mean values of three independent experiments (with 45 cells per experiment). Statistically significant differences appeared at a *p*-value of <0.001, as assessed by one-way ANOVA.

### 2.5. Sporulation Efficiency Assays

As described previously, after being pre-grown on YPD, the diploid strains were placed on a sporulation medium (0.1% yeast extract, 1% potassium acetate, 0.05% glucose and 2% agar) for 14 days at 28 °C [27]. A suspension of the cells in water was then prepared. Cells and asci were counted in a cell-counting chamber (at least 300 objects per probe). The occurrence of asci among the total cells was conveyed as the percentage of the total number of cells counted. The presented data are the mean values of the assessed numbers for three cultures per strain together with the SD.

### 2.6. Fluorescence-Activated Cell Sorting (FACS) for Cell Cycle Analyses

FACS was executed for the yeast cell cycle analysis, according to [22]. Samples were taken between the 4th and 6th h of cultivation in YPD medium at 20 min intervals, and fixed in 96% ethanol at −20 °C for at least 12 h. Then, the cells were pelleted and resuspended in sodium citrate (50 mM, pH 7) and sonicated for 20 s at 50% power. The next step was Rnase A (Merck KgaA, Darmstadt, Germany) incubation (0.5 mg/mL) for 60 min at 37 °C, monitored by staining with propidium iodide (50 µg/mL) (Sigma-Aldrich, Merck Bulgaria EAD, Sofia, Bulgaria) for 20 min in the dark.

Data acquisition was performed using the BD FACSCalibur™ Flow Cytometer (BD Biosciences, Franklin Lakes, NJ, USA). The acquired data were investigated by FlowJoV10 software (BD Biosciences, Franklin Lakes, NJ, USA). The data are represented as the mean percentage of cells in the particular cell cycle phase from the whole cell population from three independent experiments.

### 2.7. Chromatin Yeast Comet Assay (ChYCA)

The ChYCA was performed following the protocol developed by Georgieva et al. and published elsewhere [22,23]. At the two chosen time points—the 4th and 6th h of cultivation—cells aliquots were collected, followed by washing with 1xPBS (2.68 mM KCl, 1.47 mM KH_2_PO_4_, 137 mM NaCl, 8 mM Na_2_HPO_4_; pH 7) (Merck KGaA, Darmstadt, Germany) and resuspension to a concentration of 1 × 10^5^ cells/mL in S-buffer (1 M Sorbitol; 25 mM NaPO_4_, pH 6.5) (Merck KGaA, Darmstadt, Germany). After gel solidification and coverslip removal, both haploid and diploid strains were treated with 5 U/mL micrococcal nuclease (MNase; Takara Bio Inc., Kusatsu, Japan) diluted and activated in micrococcal nuclease buffer (20 mM Tris-HCl, pH 8.0; 5 mM NaCl; 25 mM CaCl_2_) and incubated for 1 min at 37 °C. The enzyme reaction was blocked by immersing the slides in a lysis solution (146 mM NaCl; 30 mM EDTA; 10 mM Tris-HCl and 0.1% N-lauroylsarcosine, pH 7.5) at 4 °C for 20 min, then washed in 0.5× TBE buffer (44.5 mM Tris, 44.5 mM Boric-acid, 2.5 mM EDTA, pH 8). Electrophoresis was conducted at 0.45 V/cm for 10 min, followed by dehydration of the slides in ethanol. The results were visualized after SYBR^®^ Green I staining using a Leitz epi-fluorescent microscope (Orthoplan, VARIO ORTHOMAT 2). Pictures were taken by a built-in microscope digital camera (Levenhuk, Inc., Tampa, FL, USA), and the parameter “Tail Moment” was analysed using the TriTek Comet Score Freeware v1.5 software (TriTek, Corp., Sumerduck, VA, USA)

The data represent mean values of the calculated parameter Tail Moment from triplicate experiments.

### 2.8. Determination of the Yeast Budding Lifespan

A micromanipulator was used to arrange the cells on a YPD plate after they were grown overnight. The budding lifespan was assessed microscopically by a routine technique, as designated before [28]. The number of daughter cells formed by a single mother cell reflected the budding lifespan (designated also as a reproductive potential). The analysis was performed by micromanipulation using the Nikon Eclipse E200 optical microscope (Nikon Instruments Inc, Melville, NY, USA) with an attached micromanipulator. The results represent measurements for at least 90 cells analysed in two independent experiments.

### 2.9. Determination of the Yeast Total Lifespan

The total lifespan of the yeast cultures was determined as previously described [29] with small modifications [30]. Then, 10 μL aliquots of a fresh yeast culture were collected and transferred to YPD plates with Phloxine B (Fisher Scientific Company, Montreal, QC, Canada) (10 μg/mL) in the medium. In each experiment, 45 single cells were analysed. During manipulation, the plates were kept at 28 °C for 15 h, and at 4 °C during the night. During the routine determination of the total lifespan, the number of crushed cells was calculated. The presented results are measurements for at least 90 cells analysed in two independent experiments.

### 2.10. Yeast Vacuole Staining

The MDY-64 (Cat. #Y7536. Invitrogen, Waltham, MA, USA) vacuole membrane marker was used as described by the manufacturer. Cells were cultivated to an OD_600_ of 0.5–0.6. Then, they were washed two times with deionized water. The next step was to resuspend the cells at 10^6^ cells/mL in 10 mM HEPES buffer pH 7.4 (Sigma-Aldrich, Taufkirchen, Germany) containing 5% glucose. The MDY-64 marker was added to a final concentration of 10 µM and cells were incubated at room temperature for 5 min. Next, the supernatants were discarded and the pellets were resuspended in fresh 10 mM HEPES buffer pH 7.4 containing 5% glucose. The fluorescence images were taken using an Olympus BX-51 microscope equipped with a DP-72 digital camera and images were quantified by the cellSens dimension software (Olympus Europa Holding GmbH, Hamburg, Germany).

### 2.11. Measurement of the Amount of Superoxide Anion Generation from the Yeast Cells during Their RLS

The amount of the produced reactive oxygen species (superoxide anion) was assessed with dihydroethidium (hydroethidine, DHET; final concentration 18.9 μM) (Invitrogen, ThermoFisher Scientific, Waltham, MA USA), as described previously [30]. Exponentially growing yeast cells were washed with sterile water and diluted to the final density of 10^8^ cells/mL in 100 mM phosphate buffer pH 7.0 containing 0.1% (*w*/*v*) glucose and 1 mM sodium EDTA. The kinetics of fluorescence increase due to the oxidation of hydroethidine was measured by the TECAN Infinite 200 microplate reader (Tecan Trading AG, Männedorf, Switzerland) at λ_ex_ = 518 nm and λ_em_ = 605 nm at 28 °C.

The data represent the mean values from triplicate measurements.

### 2.12. Polysome Profiles of the Studied Yeast Cells during the RLS

The polysomic profile analyses were accomplished by spinning down the total cell extracts in 5–50% linear sucrose gradients. Cell extracts were maintained following the protocol published elsewhere [31]. Aliquots of the lysate equivalent to A_260_ 15 units were loaded on a linear sucrose gradient and were spun down for 4.5 h at 26,500 rpm at 4 °C in a Beckman Coulter ultra-centrifuge (SW32Ti rotor) (Beckman Coulter, Inc., Brea, CA, USA). The resulting fractions were examined by an ISCO Brendel density gradient fractionator (Brandel, Gaithersburg, MD, USA).

### 2.13. Phenotypic Analysis—Spot Tests

Yeast strains were cultivated to the exponential phase (OD_600_ nm between 0.8 and 1). Subsequently, they were successively diluted as indicated. Then, 5 μL of each cell suspension was spotted onto agar plates containing different concentrations of Congo red (Sigma-Aldrich, St. Louis, MO, USA), calcofluor white (CFW; Sigma-Aldrich), sodium chloride (NaCl; Sigma-Aldrich, St. Louis, MO, USA), Camptothecin (Sigma-Aldrich, St. Louis, MO, USA), G418 (Sigma-Aldrich, St. Louis, MO, USA), Zeocin and SDS. Growth was registered 48 h after incubation at 30 °C. All defined phenotypes were established by triplicate measurements.

### 2.14. Examination of the Yeast Cells’ Chemical Composition by Raman Spectra

The analysis of the chemical composition of the yeast was performed using a Nicolet NXR 9650 FT-Raman Spectrometer (HAMAMATSU PHOTONICS Europe GmbH, Herrsching, Germany) equipped with an Nd: YAG laser 1064 nm and an InGaAs detector. Before the measurements, the yeast samples were lyophilized. FT-Raman spectra were measured in the range from 400 cm^−1^ to 2000 cm^−1^ using 64 scans with 8 cm^−1^ spectral resolution at a laser power of 0.5 W. The results are shown on the averaged spectra with 6 independent repetitions. Raman spectra were analysed by the Omnic/Thermo Scientific software (Thermo Fisher Scientific, Waltham, MA, USA), whereas baseline correction and normalization of the obtained spectra were performed using OPUS 7.0 software (Bruker Corporation, Billerica, MA, USA).

### 2.15. Statistical Analysis

All statistical analyses were performed using the Statistica 10 software. The results represent the mean values ± SD from duplicate or triplicate experiments as indicated for each experiment. One-way ANOVA and Dunnett’s post hoc tests were applied for data evaluation. The values were classed as significant at * *p* < 0.05, ** *p* < 0.01 and *** *p* < 0.001.

## 3. Results

### 3.1. The Haploid Mutants arp4 and arp4 hho1Δ Exhibit Abnormal Growth during Their RLS Due to Changes in Cell Cycle and Genome Organization

Previous reports have shown that the yeast linker histone Hho1p and actin-related protein 4 (Arp4p) interact physically with each other, which is pivotal for the maintenance of proper chromatin structure organization, genome stability and cellular morphology [20]. Recent studies have also suggested a role for these proteins in chronological ageing [32] and cellular resilience to stress during CLS [19]. So far, however, we have had little knowledge on whether and how the interactions between Hho1p and Arp4p could influence the ageing of mitotically active cells. For this purpose, we performed an extensive morphological and physiological analysis of two sets of strains—haploids (WT, *arp4*, *hho1Δ* and *arp4 hho1Δ*) and their heterozygous diploid counterparts (WT/WT, *ARP4*/*arp4*, *HHO1*/*hho1Δ* and *ARP4 HHO1*/*arp4 hho1Δ*)—to study to what extent cells need two wild-type alleles of these genes (*ARP4* is an essential gene; *HHO1* is a non-essential gene).

As shown in Figure 1a, the single *arp4* haploid mutant had the same growth rate as the wild type. However, the doubling time calculated during the routine procedure of budding lifespan determination showed that both *arp4* and the double mutant *arp4 hho1Δ* had significantly extended doubling times (*p* < 0.001) (Figure 1c). In Figure 1b, we show the growth curve assessment as the number of cells during growth. The data presented in Figure 1b confirm that *arp4* and *arp4 hho1Δ* had extended doubling time compared to the wild type. We further show that all strains had a similar colony formation ability after 48 h incubation in a fermentative (YPD) carbon source (Figure 1d).

We also explored if the lack of one functional allele of the *HHO1* and *ARP4* genes would affect the growth rate and other physiological parameters in diploid cells. For this purpose, we mated all the tested haploids of DY4285 and DY2864 (*MAT*a) with BY4742 (*MAT*α) (Table 1). Thus, we obtained heterozygotes. As shown in Figure 2 a–c, all strains had similar growth kinetics, which was confirmed by doubling time analyses. Therefore, we suggested that one wild-type allele of the *ARP4* and/or *HHO1* genes is sufficient for the cells to confer the wild-type growth phenotype in a rich fermentative medium (YPD). As shown in Figure 2d, the sporulation efficiency of diploids harbouring a point mutation in one of the *ARP4* alleles was indistinguishable from that of WT yeast cells. Previous analyses have shown that a lack of *HHO1* expression (*hho1Δ*/*hho1Δ* strain) leads to disturbed gametogenesis [33]. Interestingly, here, we showed that the presence of only one *HHO1* allele (*HHO1*/*hho1Δ* strain) leads to a significant escalation in sporulation frequency in comparison to the wild-type WT (2n) (Figure 2d). Therefore, it could be speculated that not only the availability but the amount of the Hho1p is decisive for the sporulation efficiency. As previously reported [33], a high binding correlation of Hho1 and Ume6 (the master repressor of early meiotic genes) proteins at promoters of early meiotic genes during vegetative growth was revealed by ChIP-chip. In addition, the authors showed that Hho1 and Ume6 were depleted during meiosis and Hho1p was enriched in mature spores. Authors concluded that “Hho1p may play a dual role during sporulation: Hho1 and Ume6 depletion facilitates the onset of meiosis via activation of Ume6-repressed early meiotic genes, whereas Hho1p enrichment in mature spores contributes to spore genome compaction”. The lesser amount of Hho1p in heterozygotes could facilitate and quicken the activation of early meiotic genes, but this assumption needs further detailed analyses.

Our results further proved that the combination of the two mutations (*arp4* and *hho1Δ*) in one genome (*ARP4 HHO1/arp4 hho1Δ* cells), as for the *HHO1/hho1Δ* heterozygote, regardless of the presence of a wild-type allele, did significantly affect the sporulation efficiency (*p* < 0.001). Notably, compared to *ARP4*/*arp4* and *HHO1*/*hho1Δ* heterozygotes, the heterozygous diploid double-mutant *ARP4 HHO1*/*arp4 hho1Δ* cells produced three times more and slightly fewer tetrads, respectively. This indicated that the disruption of the interaction between *ARP4* and *HHO1* impacted the sporulation efficiency and that the Arp4 protein was not completely indifferent to gametogenesis.

Next, we logically proceeded with investigating the cell cycle dynamics of the replicatively ageing haploid and diploid yeast strains. The aim was to establish whether there was a difference in the control of the cell cycle among the studied haploids and diploids having just one wild-type allele of the *ARP4* and/or *HHO1* genes. For this purpose, we monitored the cell cycle progression of both the haploid and diploid sets of the four strains (Table 1). The cells were cultivated in YPD media and aliquots were taken each 20 min between the 4th and 6th h of cultivation and prepared for FACS acquisition. The results were quantified as a percentage of cells in each phase of the cell cycle from the whole cell population accepted as 100% (Figure 3).

Normally, during exponential growth in a rich medium, the time spent in the cell cycle phases is divided into a smaller amounts of around 1/4 of the total cell population in the G0–G1 phase, compared to 3/4 being in the S-G2/M cell cycle phases [33]. Our results showed that the diploid sets of the four studied strains—WT (2n) and *ARP4/arp4*, *HHO1/hho1Δ*, and *ARP4 HHO1/arp4 hho1Δ*, bearing one wild-type allele of the *ARP4* and *HHO1* genes—represented the typical for the replicative lifespan distribution of their cells in the cell cycle phases. The majority of cells were in the S-G2/M phases for all of the studied time points.

The four haploid strains—WT (n), *arp4*, *hho1Δ*, and *arp4 hho1Δ*—showed an elongated G1 period before the majority of their cells entered S-G2/M at the 4th h and 40th min time point. Interestingly, at the 4th h and 20th min time point, the two chromatin mutants—*arp4* and *hho1Δ*—had a greater proportion of their cells in S-G2/M: 33% and 40%, respectively, compared to the WT (n), having only around 5% of cells in S-G2/M. When combining the two mutations though, this effect was lost and the G1 period was elongated further, with the majority of the double-mutant *arp4 hho1Δ* cells entering G2/M later on at the 5th h of cultivation (Figure 3). This result represents the importance of Hho1p and Arp4p in the maintenance of proper cell cycle progression during replicative ageing. On the contrary, no major differences were noted between the WT (2n) strain and the three heterozygous chromatin mutant strains, suggesting that one *ARP4* and/or *HHO1* allele was enough to confer the wild-type growth phenotype. These results were following the growth kinetics and the average doubling time detected for the WT (2n) and the three heterodiploid strains (Figure 2).

Chromatin has been proven to be a major mediator in the process of ageing [18]. During replicative ageing in yeast, the chromatin compaction has been shown to decrease, in combination with the loss of transcriptional silencing [34]. It has been found that the depletion of the H1 histone resulted in a serious stalling of the replication fork and an increase in DNA damage signalling [35]. Perturbed chromatin structure led to aggravated replication stress, causing the cells to age prematurely, as well as contributing to the development of age-associated diseases [35].

To assess changes in global chromatin compaction in the haploid and diploid sets of the four strains studied, during their replicative ageing, we performed a Chromatin Yeast Comet Assay (ChYCA). These results are the first to describe changes in chromatin among those mutants in replicatively ageing cells. Cellular chromatin was digested with micrococcal nuclease (MNase) in situ on the microscopic slides containing the microgels with lysed cells inside, followed by 10 min electrophoresis under mild conditions [19,22,23]. By doing this, the loops freed by MNase chromatin extended to the anode, forming so-called comet tails. Data were calculated for the parameter Tail Moment (TM) by the CometScore software. ChYCA was performed in three independent experiments; the results were averaged and are presented in Figure 4. In both haploid and diploid backgrounds, the double mutants had loser chromatin (higher TM), most pronounced for the haploids at the two measured time points and only for the second one for the diploids. Therefore, this suggests that one functional allele of *ARP4* and *HHO1* genes cannot compensate for these proteins in the preservation of chromatin in ageing. In more detail, the wild-type strains WT (2n) and WT (n) did not display any significant differences. The situation with *ARP4/arp4* and *arp4* strains correlated with other findings showing that the mutated *ARP4* gene caused the chromatin structure to become more condensed in comparison with the wild type [36]. The TM values for the *arp4* haploid dropped from 58 arb. u. at the 4th h to 51 arb. u. at the 6th h. For the *ARP4/arp4* heterodiploid, the TM values at the 4th h were 58 arb. u. and decreased to 50 (Figure 4). For the *hho1Δ* and *HHO1/hho1Δ* mutants, we observed subtle differences between the measured values of TM, at 61 arb. u. for the *hho1Δ* haploid and 63 arb. u. for the *HHO1/hho1Δ* heterodiploid strain at the 4th h of cultivation. These values for the haploid dropped to 48 arb. u. at the 6th h of cultivation, while for the diploid TM remained unchanged (Figure 4). The double mutants *arp4 hho1Δ* and *ARP4 HHO1/arp4 hho1Δ* at the 4th cultivation h had TM values of 61 arb. u. (haploid) and 55 (heterodiploid), respectively. These TM values increased to 68 arb u. at the 6th h for the *arp4 hho1Δ* haploid, while for the *ARP4 HHO1/arp4 hho1Δ* heterodiploid, TM values remained unchanged. Overall, with the progress of RLS, the haploid double-mutant *arp4 hho1Δ* had the highest TM, surpassing the rest of the diploid strains. These results showed that the combination of a mutation in the *ARP4* gene and the lack of Hho1p harmed chromatin structure during replicative ageing, and this was most explicitly demonstrated in the haploid cells.

### 3.2. arp4 and arp4 hho1Δ Display Accelerated Replicative Ageing, Accompanied by Alterations in the Vacuolar, Nuclear and Cell Wall Fitness and Morphology

Chromatin remodelling and DNA stability are pivotal in many biological processes, including ageing, and therefore are essential in all analyses of ageing. For that reason, in our investigation, we determined the budding lifespan of the studied haploid and diploid mutants. The budding lifespan, which is termed replicative lifespan in a standard approach, is evaluated as the number of daughter cells produced by a single mother cell. In many labs, the budding lifespan is used to determine age, yet it shows only yeast fertility or the ability to produce subsequent buds. As a physiological parameter, it is useful and determines the proliferative ability, which is the basis to use yeast as a model in cancer research or other human-related diseases. As shown in Figure 5a, both *arp4* and the double mutant had a significantly decreased budding lifespan (*p* < 0.001). In turn, *hho1Δ* was comparable to the wild type. We also showed a similar dependence in terms of reproduction lifespan (the period from the birth of the cell to the arrest of proliferation) (Figure 5b). Interestingly, as shown in Figure 5c, both *arp4* and the double mutant had significantly decreased post-reproductive lifespans (*p* < 0.001), i.e., the period between the last budding and cell death.

The post-reproductive lifespan is another crucial parameter in yeast ageing, showing clearly that cells do not die after reproduction, as previously thought. In turn, the total lifespan is defined as the sum of the reproductive and post-reproductive lifespans and determines the length of cell life.

Our studies of the reproductive and post-reproductive lifespans of the studied yeast cells showed that both *arp4* and *arp4 hho1Δ* haploid mutant strains had a significantly shorter average and maximum length of life (*p* < 0.001) (Figure 5d). Thus, both *arp4* and the double mutant had decreased budding and total lifespans leading to premature replicative ageing and death.

We then checked the ageing parameters in the diploid cohort of yeast cells. Interestingly, the appearance of one wild-type allele of the gene in heterozygous cells led to the complementation of the mutation (Figure 6a–d). This allowed us to conclude that both Arp4p and Hho1p are key players that regulate not only DNA stability but also ageing. As shown in Figure 5c, *arp4* and *arp4 hho1Δ* had a significantly decreased post-reproductive lifespan. Cells from these strains died shortly after the end of reproduction. Therefore, we proceeded with checking the main reasons for these phenotypes. Our preliminary observations showed that cells became hypertrophic and died shortly after the last budding. The number of cells in the haploid and heterozygous sets that were crushed was assessed. These results are the first to report that cells from the *arp4* and *arp4 hho1Δ* mutants burst at the end of reproduction with 70% and 90%, respectively (Figure 7a), versus 58% in the WT (n). These differences were statistically significant only for the double mutant. We assume that this cellular disaster was probably associated with the disruption of the cell membrane and the cell wall. As shown in Figure 7b, the heterozygous cells showed a much lower level of cell breakage, and this was comparable in all mutants as opposed to the wild type.

The next sets of experiments aimed to investigate the physiological causes of the loss of cell physical integrity, leading to cell death and restricting reproduction in the studied double-mutant yeast cells. Vacuole morphology is accepted as a biomarker for ageing [37]. Previous results showed that the haploid *arp4*, *hho1Δ* and *arp4 hho1Δ* mutant cells had—at the early time points of their CLS—prematurely aged and damaged vacuoles, and this tendency was most pronounced for the double-mutant cells [31]. For this purpose, we started by checking the morphology of the vacuoles. As shown in Figure 8a, both *arp4* and *arp4 hho1Δ* had a single wrinkled vacuole, while the wild type and *hho1Δ* usually had two or more. It is, therefore, suggested that the cause of cell rupture could be vacuole fusion, as proposed earlier [38]. Interestingly, the heterozygous cohorts of yeast cells had several vacuoles in the cell, which appeared with normal morphology. This, we assume, is a biophysical advantage over the haploids (Figure 8b).

As reported earlier, the fusion of the vacuole alone is most likely insufficient, but most often cell rupture is also accompanied by disturbances in the biosynthesis of the cell wall [39]. Therefore, to strengthen our hypothesis, we performed cell wall toxicity tests on the analysed cells. Our research clearly showed that *arp4* and *arp4 hho1Δ* haploid mutants had dysfunctions related to the cell wall and cell membrane biosynthesis. As shown in Figure 9, both mutants were sensitive to cell wall biosynthesis inhibitors such as calcofluor white and Congo red with the double mutant appearing more sensitive. These dependencies were not observed in parallel comparisons with the heterozygous sets (Figure 10), although the reduction in the growth of *ARP4 HHO1*/*arp4 hho1Δ* heterodiploid on Congo red and SDS, as well as on NaCl, was not as explicit as for the haploid double mutant (Figure 9). Furthermore, haploids *arp4* and the double mutant were also more sensitive to osmotic stress (growth with NaCl, more discernible for the double mutant), oxidative stress, and G418, which, as a ribosome inhibitor, blocked the polypeptide synthesis by inhibiting the elongation step (Figure 9). In the heterozygous set, only *ARP4/arp4* differed in growth on the G418, while the double mutant showed resistance to G418 comparable to that of the wild type. This again demonstrates the effect of the simultaneous presence of both mutations in one genome and the influence of hho1 deletion on the double mutant phenotype (Figure 10). The toxicological analyses also showed that mutations in the studied genes did not affect the effectiveness of DNA repair after treatment with zeocin (data not shown). Interestingly, Camptothecin, a DNA topoisomerase I inhibitor, significantly inhibited the growth of *arp4* and *arp4 hho1Δ*. Overall, as expected, the haploid strains were more sensitive to the applied treatments than the heterodiploid counterparts (Figure 9 vs. Figure 10). The haploid double-mutant *arp4 hho1Δ* displayed higher sensitivity to Congo red, CFW (25 μg/mL), NaCl and Camptothecin than the *arp4* that reflects the influence of *HHO1* deletion. Additionally, *arp4 hho1Δ* was more sensitive to G418 (50 μg/mL), in comparison to the WT and *hho1Δ* strains, but more resistant compared to the *arp4* mutant (Figure 9), an indication that Hho1p depletion is suppressive to *arp4* mutation.

#### Translational Fitness

Our research shows that G418 is toxic to *arp4* mutant cells, indicating a biological link with the ribosome biosynthesis in these cells. For this purpose, the polysome profile was determined. The metabolic fitness of the analysed yeast strains was additionally evaluated by examination of the translational apparatus performance, as translational machinery can be regarded as a sensor of metabolic perturbation. Under steady-state conditions, in logarithmically growing yeast cells, translation consumes the majority of cellular energy resources, and thus perturbations in many metabolic circuits usually affect protein synthesis. The best approach to evaluate translational fitness is to analyse the polysome profiles. The analysis is considered a very profound approach to detect various defects of translation, and at the same time cellular metabolism. This method permits the characterization of translational machinery through studying the engagement of particular ribosomal elements in the translational process by following the content of the free ribosomal subunits 40S and 60S, as well as 80S (considered as transnationally active monosomes and vacant ribosomes, disengaged from translation), and polysomes considered as multiple 80S on a single mRNA [39].

We performed polysome profile analysis for all tested strains (Figure 11). In the case of the heterodiploid strains with a single mutant gene or double mutants, the polysome profiles (the calculated 40S/60S and polysome/monosome ratio—P/M) differed slightly from the wild-type strain with little differences. There was a 10% reduction in the P/M ratio in *HHO1/hho1Δ* and *ARP4/arp4,* and a 10% increase in the P/M ratio in *ARP4 HHO1/arp4 hho1Δ* in comparison with the WT (2n): 3.45, 3.47 and 4.39 vs. 3.95, respectively. The detected differences between the single and double mutants were more than 25%: 3.47 vs. 4.39. This finding indicated that the lack of one gene or both in the diploid genetic background slightly affected the metabolic fitness of the strains, as measured by the polysome profile analysis. In the case of the haploid strains, the effect of the single gene mutation (*ARP4*) or the deletion (*HHO1*) was more pronounced. In the case of the two haploid *arp4* and *hho1Δ* mutants, the analysis showed clear perturbations in translation. These perturbations involved an increase in the monosomes to polysomes ratio. This was an indication that, in both strains, the accumulation of 80S ribosomal particles was taking place, suggesting that the initiation of translation and/or recycling of the vacant 80S might be perturbed. On the other hand, the 40S/60S ratio was almost unchanged, indicating that the biogenesis of the ribosomal subunits was not affected.

An interesting effect was observed for the double-mutant haploid strain, *arp4 hho1Δ*, where the polysome profile was different from the WT (n), with a P/M ratio 10% higher than that of the WT (n), but with a 40S/60S ratio similar to the WT (n). The observed effect indicated that abolishing the native function and/or the interaction of the two proteins (Arp4p and Hho1p) provided a positive synergistic effect, restoring the performance of the translational machinery, and in the mutant *arp4 hho1Δ*, the lack of both proteins bypassed the deficiency of the single protein, which exerted a negative effect. Thus, the polysome profile analysis showed that the proteins Arp4p and Hho1p displayed functional links, supporting previous analyses which showed that both proteins were reported to be chromatin components and proteins that physically interact with each other while maintaining important functions such as stress resilience, chromatin dynamics and replication in CLS [20,31]. The results of this study prove their concomitant involvement in RLS too.

Oxidative stress plays a pivotal role in the ageing of all organisms, including yeast [40]. Therefore, we went on to determine the superoxide anion level in cells treated with hydrogen peroxide. As shown in Figure 12a, all mutants had a significantly increased superoxide anion level under control conditions (without H_2_O_2_) compared to the reference strain, WT (n). In turn, the reactive oxygen species (ROS) levels in cells exposed to oxidative stress were significantly increased in the case of *arp4* and the double mutant, at *p* < 0.001 and *p* < 0.05, respectively. Interestingly, in the case of the diploid set, only the wild-type strain had an elevated ROS level upon H_2_O_2_ treatment, and all heterozygous strains had a significantly reduced level of ROS compared to that of the WT (2n) (*p* < 0.001). Importantly, in all strains, haploids and diploids, the level of ROS was higher after oxidative stress.

### 3.3. Changes in the Chemical Composition of the Haploid and Heterozygous Strains

The alterations in the chemical composition of the wild-type cells and the mutants of haploid and heterozygous yeast cells were investigated by the analysis of the Raman spectra and the results are presented in Figure 13. In Table 2, the position of the peaks and their shifts corresponding to vibrations of functional groups are presented for all samples of interest. Strains harbouring any of the mutant genes in both haploids and diploids are characterized by shifted bands (Raman shift) compared to the corresponding WT. The spectra of the WT (n) and mutant haploid cells were not similar and had visible differences (Figure 13a). The haploid yeast strains with a single mutation (*arp4* and *hho1Δ*) had similar chemical compositions, while the cells’ spectra had similar bands of similar intensity corresponding to RNA (719, 1005, 1097, 1325 cm^−1^), proteins (1005, 1606 and 1660 cm^−1^), polysaccharides (426, 900, 966, 1097, 1151 cm^−1^) and lipids (1255 and 1660 cm^−1^). The spectra of the other two haploid strains (WT (n) and *arp4 hho1Δ*) had similar bands but with lower intensity compared to that of the *arp4* and *hho1Δ* strains, indicating a lower concentration of chemical compounds (Figure 13a). Additionally, the spectrum of the *hho1Δ* strain was characterized by bands of larger intensity corresponding to RNA (51–550 cm^−1^) and polysaccharides (950 cm^−1^) in comparison to other studied haploid cells. For the *arp4* and *arp4 hho1Δ* haploids, no amine II peak was found at 1520–1535 cm^−1^, which indicated differences in RNA/DNA for these strains compared with the WT (n) and *hho1Δ* strains. The presence of the band that corresponded to proteins and mitochondria (1610 cm^−1^) was found in the spectra of the haploid mutants. This difference indicated changes in the protein and mitochondrial composition in the mutant strains. Moreover, the spectrum of the *arp4 hho1Δ* haploid cells had the lowest intensities of the bands, indicating a much slower growth rate (as shown in Figure 1 and Figure 5) of these cells, or a slower metabolism.

The spectra of diploid cells were characterized by a higher intensity of the bands compared to the spectra of haploid cells, especially for the WT and double mutant (Figure 13b). This may suggest a higher concentration of chemical compounds in the diploid cells in comparison to haploid cells. Except for some small differences, the spectra of WT (2n) and heterodiploid mutants were similar to each other in the whole measurement range, which suggested similar chemical composition. One of the mentioned differences was the absence of a protein band (854–858 cm^−1^) for the *HHO1/hho1Δ* strain (Figure 13b). This shows variability in the protein composition of these cells in comparison to the WT (2n). Additionally, the spectra of diploid mutants had additional bands corresponding to RNA/DNA (610–620 and 1370–1395 cm^−1^) and polysaccharides (418–430 cm^−1^), which were absent in the spectrum of the WT (2n) cells. This indicated differences in the RNA/DNA and polysaccharide composition of the mutant cells compared to the WT (2n) cells. The band in the 757–761 cm^−1^ range (proteins/Cyt c) appeared only in the spectra of diploid cells, which distinguished them from the haploid cells (compare Figure 13a,b and Table 2).

## 4. Discussion

Recent studies of several model organisms have indicated chromatin structure and its remodelling as an important player in ageing [10,18,41]. During yeast replicative ageing, there is a loss in the silencing of the rDNA region, an increased formation of extra ribosomal chromosomal circles, and an increase in genomic instability. An interesting protein that acts in aged yeast and is involved in chromatin remodelling is the silencing information regulator 2 (Sir2p) [42,43]. The deletion of *SIR2* decreases the budding lifespan, while the overexpression of *SIR2* increases it [44]. Sir2p acts as a deacetylase of histone H4 lysine 16 (H4K16Ac) [45]. It is interesting that during yeast ageing, the level of Sir2 proteins decreases, allowing for an increase in the H4K16Ac marker [43]. As shown previously, during yeast ageing, variations in gene expression patterns are also observed [46,47], mainly due to stress responses, gene repair and altered chromatin structure ageing. Previous data have proven that the structure of chromatin plays an important role in ageing [21]. Recently, much attention has also been paid to mutations in the genes *HHO1* and *ARP4* coding for the yeast H1 linker histone and the actin-related protein 4, respectively. The latest data have shown that the budding yeast Hho1p physically interacts with Arp4p, and that the abolishment of this interaction leads to global changes in chromatin compaction [20] and, therefore, is critical for maintaining genome stability, stress response, and ageing [32]. Miloshev et al. (2019) suggested that linker histones, chromatin remodelling and the Arp4p and Hho1p interaction are crucial in maintaining genome stability and chronological ageing. Hence, the question that arises is about the effect of *arp4* and *hho1Δ* mutations on the ageing of mitotically active haploids and heterozygote diploids. Our preliminary studies clearly showed that the *arp4* mutant and the double mutant had an abnormal growth rate, as evidenced by the doubling time conversion. Interestingly, the availability of one of the wild-type alleles of the *ARP4* gene was sufficient to confer the growth rate of heterozygotes to that of the WT (2n). This most likely means that there was no association with the number of the *ARP4* gene alleles. A recent report showed that limiting the number of mRNA transcripts in a cell in a heterozygous system associated with replication initiation was crucial to achieving the extension of cell doubling time [48]. This report used *arp4* as a conditional mutant. It has G1/S phase suspension in the cell cycle [49] with additionally elevated spindle elongation, breakage associated with lagging chromosomes in mother cells, and abnormal nuclear morphology. Physical interactions between Arp4p and histones in vitro [50,51] and the association of Arp4p with chromatin in vivo have been demonstrated previously [24]. Thus, significantly extended doubling time is strongly associated with cell cycle disorders and probably DNA damage. The inability to repair the DNA double-strand break (DSB) can be disastrous. In turn, survived cells may display aneuploidy or other chromosome defects, which may ultimately lead to cancer [52,53]. In a properly functioning eukaryotic cell, DNA damage foci (as a response to DSB) lead to a significant reorganization of chromatin structure [54,55]. Recent data suggest that Hho1p is compulsory for efficient sporulation and plays a main function in the progression of meiosis. The *hho1*Δ/*hho1*Δ yeast displays a reduction in sporulation efficiency compared to the wild type [32]. However, we report here that the heterozygous strain *HHO1/hho1Δ* significantly increased sporulation efficiency. Therefore, we show the important contribution of the Hho1p protein to sporulation, but our data also suggest that the number of *HHO1* mRNA transcripts may determine sporulation efficiency.

Ageing is an inevitable biological process [56]. Ageing research conducted on model organisms such as yeast, fruit flies or worms has demonstrated that there are genes that alter the rate of ageing [39,57,58,59]. Here, we are the first to show that the list of genes that accelerate ageing should be expanded to include *ARP4* and *HHO1*. The combination of *ARP4* mutation with the deletion of *HHO1* induces a significant shortening of both yeast budding and total lifespan. These data may confirm the general statement of evolutionary biologists that organisms are programmed for survival and not for death. We hypothesize here that this premature ageing phenotype is closely correlated with the additional functions of the Arp4 protein. We suggest that Arp4p, together with Hho1p, may be involved in the molecular pathways of cell wall biosynthesis in response to environmental stress. Thus, the two proteins Arp4p and Hho1p play important roles in the adaptation of yeast cultures to varying environments.

Our results show that Arp4p disturbances in the haploid system (*arp4* and *arp4 hho1Δ*), but not in the heterozygote system, lead to increased sensitivity to inhibitors of protein biosynthesis. This is accompanied by cell rupture with an extremely short lifetime after reproduction. Here, we support the hypothesis that disturbances in cell wall biosynthesis and vacuole fusion are key factors in the determination of the budding and total lifespans [28,39]. Our research also shows that biophysical changes due to the pressure of a single vacuole on the cell wall have a drastic impact on survival when problems with remodelling or cell wall biosynthesis occur, as observed in *arp4* and *arp4 hho1Δ*. We also show here that a disruption in Arp4p function leads to the generation of more ROS. Despite the controversy in the free radical theory of ageing [60], oxygen-free radicals may also be an additional factor accelerating ageing. Nevertheless, heterozygotes displayed lower levels of ROS than wild-type strains. Recent studies evaluated oxidative stress adaptation mechanisms after deleting genes required for growth. It has been found that gene loss can enhance an organism’s ability to evolve and adapt. According to these findings, the loss of a single gene can facilitate adaptation through the opening of alternative evolutionary pathways [61].

To summarize, we have shown here that the actin-related protein 4 (Arp4p) is a factor that, by its interactions with the linker histone (Hho1p), controls the ageing of mitotically active yeast cells. We hypothesize that the *arp4* mutant and the double mutant significantly shorten both total lifespan and reproductive potential. We suggest that Arp4p and its interaction with Hho1p influence the regulation of the cell cycle, cell wall biosynthesis and cell response to environmental stress, including osmotic and oxidative stress. We also report that a lack of the H1 linker histone does not affect replicative ageing independently. In the heterozygote system, it boldly increases the efficiency of sporulation, which is why it contributes significantly to gametogenesis. Notably, we show for the first time the lack of changes in the studied ageing hallmarks in the heterozygous system, showing that one allele of the gene is sufficient to maintain the wild-type phenotype in the set-up of *ARP4/arp4*, *HHO1/hho1Δ* and *ARP4 HHO1/arp4 hho1Δ.* Interestingly, our recent data published elsewhere showed that cells with only one allele of the *ORC* genes (essential genes) had an important decline in *ORC* mRNA levels, a suspension in the G1 phase of the cell cycle, and a prolonged doubling time. We also showed that reducing Orc1-6 protein levels considerably prolonged both the budding and average chronological lifespans [48]. These results provide an elegant system and a know-how for the in-depth study of gene interactions in haploid and diploid backgrounds, thus providing a tool for assessing the weight of a mutation on ageing and the other cellular functions associated with it.

## Figures and Tables

**Figure 1 cells-11-02754-f001:**
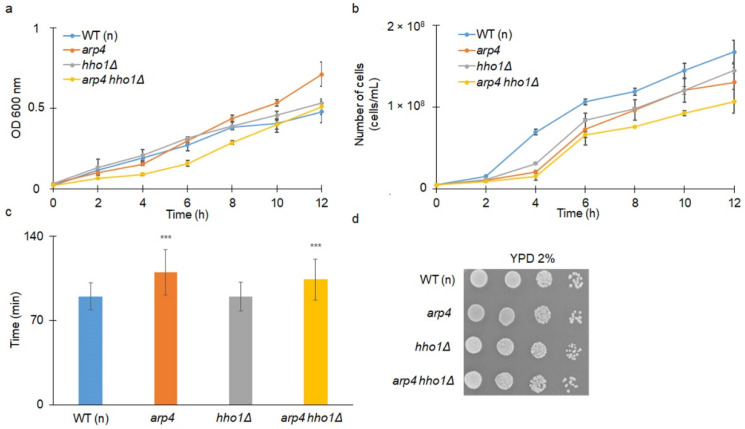
(**a**,**b**) Comparison of the yeast cells’ growth kinetics; (**c**) average doubling time during reproduction; and (**d**) growth on solid YPD after 48 h incubation at 28 °C of the haploid wild-type cells and the isogenic strains *arp4*, *hho1Δ* and *arp4 hho1Δ*. The optical density (OD_600_ nm) was measured at different time points for up to 12 h. Data are mean vales plus SD from triplicate experiments. ANOVA and the Dunnett’s post hoc tests were performed and statistically significant differences are designated at *** *p* < 0.001, compared to the control wild type (WT (n)).

**Figure 2 cells-11-02754-f002:**
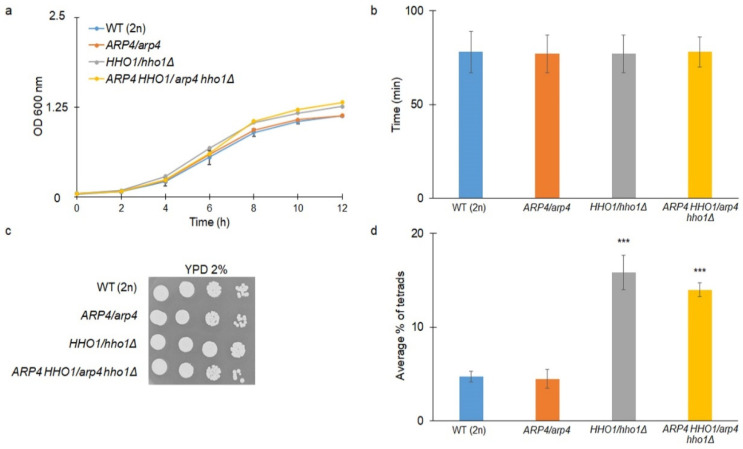
(**a**) Comparison of the yeast strains’ growth kinetics; (**b**) average doubling time during reproducing; and (**c**) growth on YPD plates after 48 h incubation at 28°C of the diploid wild-type yeast strain and its isogenic heterozygous strains *ARP4*/*arp4*, *HHO1/hho1Δ* and *ARP4 HHO1/arp4 hho1Δ.* OD_600_ of the cultures was evaluated at different time points for 12 h. (**d**) Sporulation frequency of the wild-type and isogenic heterozygous strains. The presented data are mean values with SD obtained from triplicate experiments. Statistical significance was evaluated by one-way ANOVA and the Dunnett’s post hoc tests, where *** *p* < 0.001.

**Figure 3 cells-11-02754-f003:**
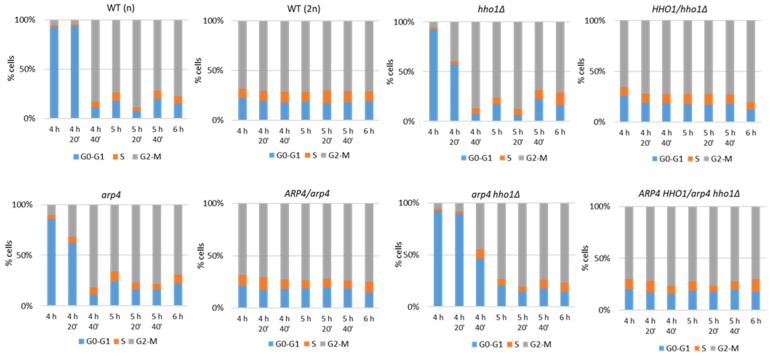
Cell cycle analysis of replicatively ageing haploid and diploid wild-type yeast strains and their isogenic mutant strains. Cells were grown in YPD for 6 h. At seven time points every 20 min between the 4th and 6th h, aliquots were taken from the cell cultures and analysed by FACS after propidium iodide staining.

**Figure 4 cells-11-02754-f004:**
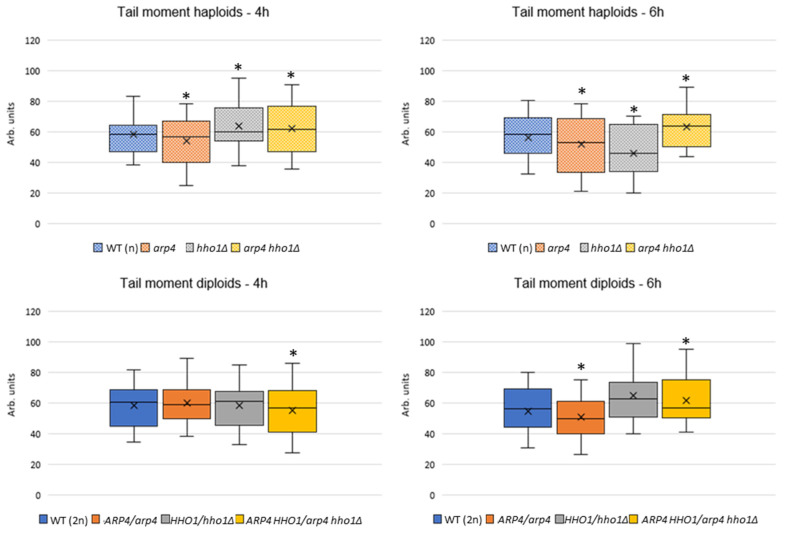
Global chromatin organization in the studied haploid and diploid WT and chromatin mutant strains assessed by Chromatin Yeast Comet Assay (ChYCA). Cells were cultivated for 6 h. At two time points (the 4th and 6th (RLS) h) aliquots of cell cultures were taken and analysed by ChYCA. The Tail Moment parameter was measured with the CometScore software. Statistical significances were assessed using ANOVA and Dunnett’s post hoc tests (* *p* < 0.05). Data represent the box–whisker plots from three independent experiments.

**Figure 5 cells-11-02754-f005:**
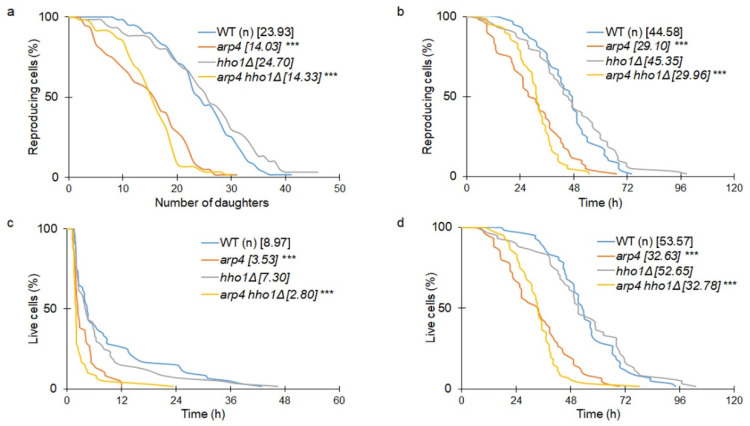
(**a**) Comparison of the budding lifespan among the studied yeast strains; (**b**) reproductive lifespan; (**c**) post-reproductive lifespan; and (**d**) total lifespan of the haploid wild-type yeast strain and its isogenic mutant strains *arp4*, *hho1Δ* and *arp4 hho1Δ.* ANOVA and Dunnett’s post hoc tests were applied and *** *p* values < 0.001 were considered significant. Data represent mean values from two independent experiments. The mean values for 80 cells from two independent experiments are displayed in parentheses.

**Figure 6 cells-11-02754-f006:**
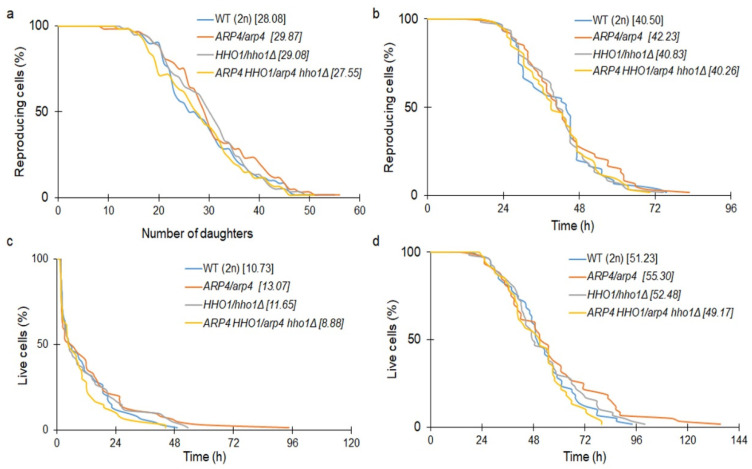
(**a**) Comparison of the budding lifespan among the studied yeast strains; (**b**) reproductive lifespan; (**c**) post-reproductive lifespan; and (**d**) total lifespan of the diploid wild-type yeast strain and its isogenic heterozygous strains *ARP4*/*arp4*, *HHO1/hho1Δ* and *ARP4 HHO1/arp4 hho1Δ.* ANOVA and Dunnett’s post hoc tests were applied and represented data are the mean values from two independent experiments. The mean values for the total 80 cells from two independent experiments are given in parentheses.

**Figure 7 cells-11-02754-f007:**
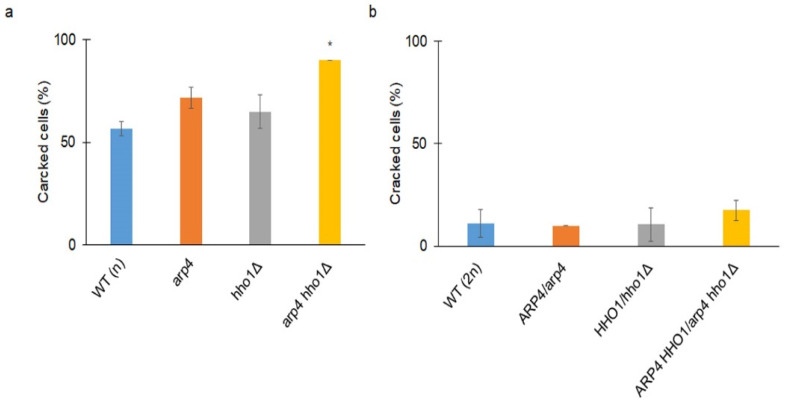
(**a**) The percent of cells that died by cell crack during the budding lifespan analyses of the haploid wild-type yeast strain and isogenic mutant strains *arp4*, *hho1Δ* and *arp4 hho1Δ* and (**b**) of the diploid wild-type yeast strain and isogenic strains *ARP4**/arp4*, *HHO1/hho1Δ* and *ARP4 HHO1/arp4 hho1Δ*. Statistical significances were assessed using ANOVA and Dunnett’s post hoc tests (* *p* < 0.05). Data represent mean values from three independent experiments.

**Figure 8 cells-11-02754-f008:**
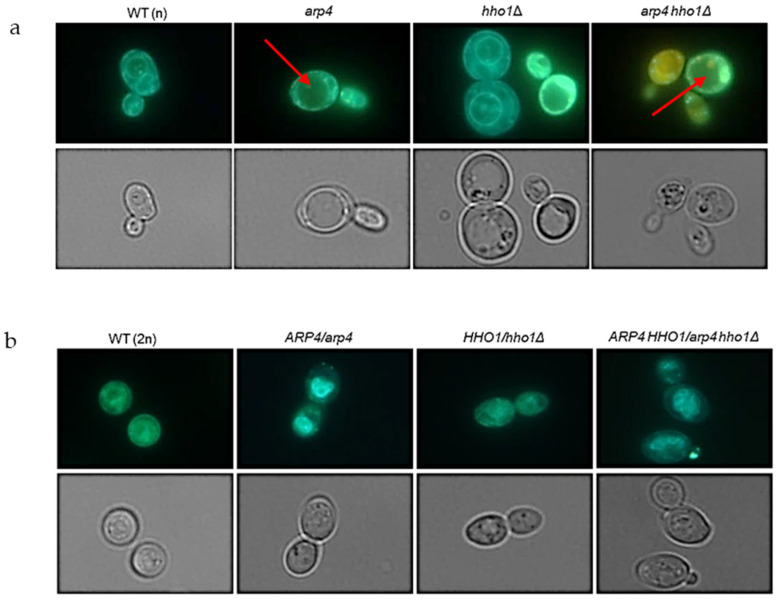
(**a**) Vacuole morphology analysis for the haploid wild-type yeast strain and isogenic mutant strains *arp4, hho1Δ and arp4 hho1Δ,* and (**b**) diploids WT (2n), *ARP4*/*arp4, HHO1/hho1Δ* and *ARP4 HHO1/arp4 hho1Δ*. Cells, cultivated in YPD to an early log phase, were stained using the MDY-64 vacuolar marker. Fluorescence pictures were taken with an Olympus BX-51 microscope equipped with a DP-72 digital camera. The cellSens dimension software was applied for image data quantitation. The red arrows in the figure indicate large vacuoles (1000× magnification).

**Figure 9 cells-11-02754-f009:**
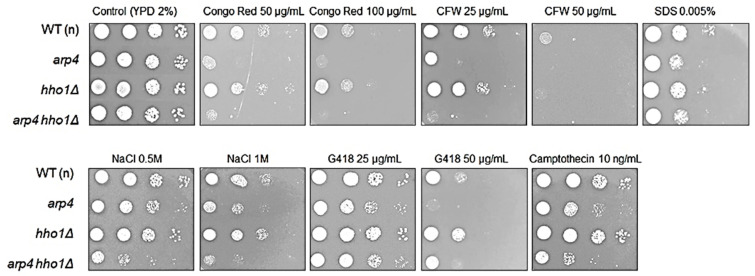
Sensitivity of the haploid wild-type yeast strain and its isogenic mutant strains *arp4, hho1Δ* and *arp4 hho1Δ* to different stress factors, designated at the top of the micrographs. Cells were grown in liquid YPD medium, counted and diluted serially to obtain suspensions at densities of 1,000,000, 100,000, 10,000 and 1000 cells. Plates were hatched for 2 days at 28 °C.

**Figure 10 cells-11-02754-f010:**
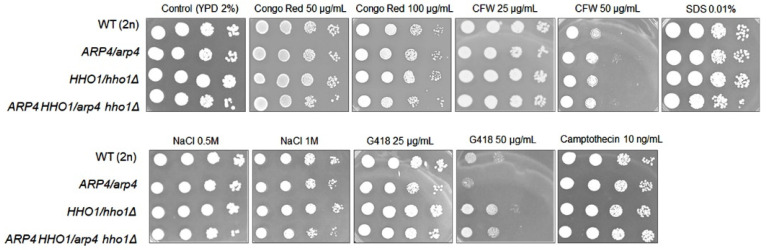
Sensitivity of the diploid wild-type yeast strain and isogenic heterodiploid strains *ARP4*/*arp4, HHO1/hho1Δ* and *ARP4 HHO1/arp4 hho1Δ* on cells’ stress factors. Cells were cultured overnight in liquid YPD medium, counted and diluted serially to obtain suspensions at densities of 1,000,000, 100,000, 10,000 and 1000 cells. Plates were incubated at 28 °C for 2 days.

**Figure 11 cells-11-02754-f011:**
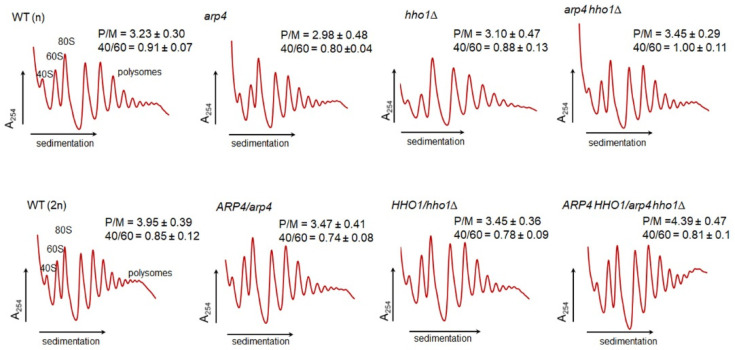
Polysomic profile analysis. The polysome profiles attained for the diploid wild-type yeast strain and isogenic heterozygous strains *ARP4*/*arp4, HHO1/hho1Δ* and *ARP4 HHO1/arp4 hho1Δ* for the haploid wild-type yeast strain and isogenic mutant strains *arp4, hho1Δ* and *arp4 hho1Δ.* The sedimentation vector of the ribosomal fractions is designated by a horizontal arrow. The optical density analysis at 254 nm is shown on the y-axis. The position of individual ribosomal subunits is indicated. Insets: P/M—the polysomes to monosomes ratio; 40S/60S—ratio of small to large ribosomal subunits.

**Figure 12 cells-11-02754-f012:**
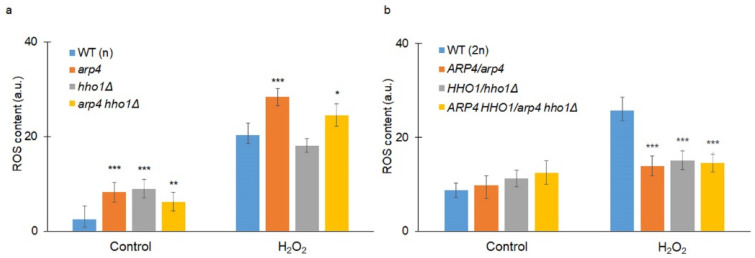
(**a**) Superoxide anion (one type of ROS) content was estimated with the fluorescent probe dihydroethdium (hydroethidine) for the haploid wild-type yeast strain and its isogenic mutant strains *arp4*, *hho1Δ* and *arp4 hho1Δ,* and (**b**) for the diploid wild-type yeast strain and isogenic heterozygous strains *ARP4*/*arp4*, *HHO1/hho1Δ* and *ARP4 HHO1/arp4 hho1Δ*. Yeast cells were grown on liquid medium without treatment (control) and treated with 1mM H_2_O_2_ for 1 h. ANOVA and Dunnet post hoc tests were performed. Data are mean values ± SD from triplicate experiments. Bars indicate SD; * *p* < 0.05 ** *p* < 0.01 and *** *p* < 0.001 designate significant differences compared to the control.

**Figure 13 cells-11-02754-f013:**
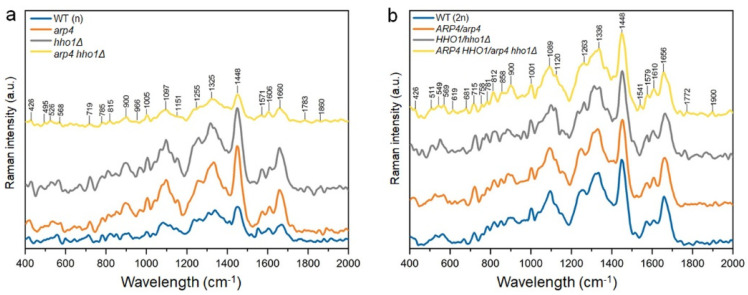
Raman spectra of the haploid (**a**) and diploid (**b**) yeast strains with characteristic peaks describing the chemical composition of cells.

**Table 1 cells-11-02754-t001:** Genotypes of the studied yeast *S. cerevisiae* strains.

Strain	Appears in Text	Genotype	Source
Haploids			
DY2864	WT (n)	*MAT*a *his4-912Δ-ADE2 his4-912Δ lys2-128Δ can1 trp1 ura3 ACT3*	[24]
DY4285	*arp4*	*MAT*a *his4-912Δ-ADE2 lys2-128Δ can1 leu2 trp1 ura3 act3-ts26*	[24]
DY2864 *hho1Δ*	*hho1Δ*	*MAT*a *his4-912Δ-ADE2 his4-912Δ lys2-128Δ can1 trp1 ura3 ACT3 ypl127C::K.L.URA3*	[20]
DY4285 *hho1Δ*	*arp4 hho1Δ*	*MAT*a *his4-912Δ-ADE2 lys2-128Δ can1 leu2 trp1 ura3 act3-ts26 ypl127C::K.L.URA3*	[25]
Diploids			
BY4742/DY2864	WT (2n)	*MAT*α/a *ADE2*/*his4-912Δ-ADE2 his3-Δ1/his4-912Δ leu2Δ0/LEU2 lys2Δ0/lys2-128Δ CAN1/can1 TRP1/trp1 ura3-Δ0*/*ura3 ACT3/ACT3*	this study
BY4742/DY4285	*ARP4/arp4*	*MAT*α/a *ADE2/**his4-912Δ-ADE2 lys2Δ0/lys2-128Δ CAN1/can1 leu2Δ0/leu2 TRP1/trp1 ura3-Δ0*/*ura3 ACT3/act3-ts26 (ARP4/arp4-ts26)*	this study
BY4742/DY2864 *hho1Δ*	*HHO1/hho1Δ*	*MAT*α/a *ADE2/his4-912Δ-ADE2 his3-Δ1/his4-912Δ leu2Δ0/LEU2 lys2Δ0/lys2-128Δ CAN1/can1 TRP1/trp1 ura3-Δ0/ura3 ACT3/ACT3 YPL127C/ypl127C::K.L.URA3 (HHO1/hho1Δ)*	this study
BY4742/DY4285 *hho1Δ*	*ARP4 HHO1/arp4 hho1Δ*	*MAT*α/a *ADE2/**his4-912Δ-ADE2 lys2Δ0/lys2-128Δ CAN1/can1 leu2Δ0/leu2 TRP1/trp1 ura3-Δ0/ura3 ACT3/act3-ts26 YPL127C/ypl127C::K.L.URA3 (HHO1//hho1Δ)*	this study

**Table 2 cells-11-02754-t002:** The Raman peaks and their shift for the analysed yeast lines with the description of vibrations consistent with the respective functional groups.

Diploids				Haploids				
WT (2n)	*ARP4*/*arp4*	*HHO1*/*hho1Δ*	*ARP4 HHO1*/*arp4 hho1Δ*	WT (n)	*arp4*	*hho1Δ*	*arp4 hho1Δ*	Vibrations
Peaks Position (Raman Shift, cm^−1^)								
1660	1660	1664	1656	1664	1660	1660	1660	Lipids, Proteins (2, 4–6)
1610	1606	1606	1610	-	1610	1610	1606	Mitochondria, Proteins (5, 7, 8)
1571	1575	1575	1579	1579	1575	1572	1572	RNA/DNA (9, 11)
1514	1525	-	1540	1521	-	1533	-	Amide II (11)
1448	1452	1452	1448	1448	1448	1448	1452	Proteins (5, 6)
-	1394	1378	1371	-	1390	-	1394	RNA/DNA (9, 11)
1332	1332	1340	1336	1340	1332	1336	1324	RNA (3, 9, 10)
1255	1247	1263	1263	1274		1259	1266	Lipids (1, 4)
1097	1093	1101	1089	1082	1097	1097	1097	Polysaccharides, RNA/DNA (3,5,6)
1001	1001	1001	1001	1005	1005	1005	1005	Proteins, Phenylalanine, RNA (4, 6, 8)
966	950	966	958	962	969	966	939	Polysaccharides (1–3)
900	889	897	897	912	897	897	900	Polysaccharides (1, 2, 6)
854	854	-	858	854	850	846	854	Proteins (5, 6, 8)
819	819	823	815	823	811	807	819	RNA, Proteins (11)
784	784	781	784	785	781	781	785	RNA/DNA (9, 11)
757	761	761	757	-	-	-	-	Proteins, Cyt c (11, 12)
719	723	723	715	711	719	719	719	RNA (3, 9, 10)
669	669	665	680	669	669	669	649	RNA/DNA (3, 9, 11)
-	622	611	615	626	611	623	623	
564	568	549	541	557	572	564	568	
526	526	514	510	530	537	526	526	RNA, Polysaccharides (3, 10)
-	418	429	426	422	422	430	426	Polysaccharides (1, 2, 6)

## Data Availability

Not applicable.

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
