# Peer review of "Actin-Related Protein 4 and Linker Histone Sustain Yeast Replicative Ageing"

_cells, 2022, doi:10.3390/cells11172754_

Round 1
Reviewer 1 Report
This paper by Molon et al, the authors addressed the influence of Hho1p/Arp4p interaction in replicative ageing of yeast cells. By examining chromatin structure, proliferative and reproductive potential and resilience to stress in both haploid and diploid strains carrying mutations in Hho1p and Arp4 coding genes, the authors concluded that the Hho1p/Arp4p interaction could play and important role in yeast replicative ageing and sporulation efficiency.
Despite the large number of qualitative observations included in this manuscript, I feel that this work still lacks robust support to make the evidence of an involvement of Hho1p and Arp4 in RLS more compelling and convincing, beyond the ChYCA. My major concern is that the association between the Hho1p/Arp4p function in chromatin maintenance and ageing is not addressed convincingly, but it only remains as an expected evidence giving that the role of both proteins in chromatin have been already described (in some case by the same authors) and the link between chromatin perturbation and ageing is well known. The ChYCA is not enough and different additional questions need to be addressed by the authors such as whether the Hho1p/Arp4p-mediated chromatin compaction affects histone modifications, gene transcription or recruitment of ageing associated factors during RLS, just to cite some.
Additional points that need to be revised concern the interpretation of the genetic analyses at different levels.
1. In the description of strains genotype the authors should make clear to the readers how the DY4285 and DY2864 genotypes refer to arp4 and hho1D mutants since both mutations do not appear in the corresponding genotypes. I personally had to look at the reference to find out that act3 corresponds to arp4 and yll127C to hho, but this information should be provided in this manuscript
2. In comparing the phenotypes between the mutant haploid and heterozygous diploid strains, in most cases the authors claim that since the heterozygotes are wild-type that “one wild type copy…is sufficient to revert to the wild-type phenotype” (i.e Lines 287, 325 ecc…). These conclusions are misleading because they do not imply an underlying genetic interaction (as the reversion/phenotype rescue should do) but only the obvious assumption that the mutations are recessive. The authors should revise it carefully
3. There are different points in the manuscript where the authors describe that either single or double heterozygotes show growth, replication parameters (i.e line 290) as well polisome profiles (line 466) different (in some case even better) than wild-type diploid strains. To the best of my knowledge, these observations are counterintuitive and deserve an accurate description/explanation.
Minor points:
1. Images of ChYCA should be provided either in the text or in Supplementary materials
2. References 20 and 26 are identical.
Author Response
Review form 1
Thank you very much for the detailed review of our work!
It indeed helped us a lot in improving it. Moreover, it further granted us the possibility to improve our work.
We tried to address all questions and suggestions. Please, see below:
- My major concern is that the association between the Hho1p/Arp4p function in chromatin maintenance and ageing is not addressed convincingly, but it only remains as expected evidence given that the role of both proteins in chromatin have been already described (in some case by the same authors) and the link between chromatin perturbation and ageing is well known. The ChYCA is not enough and different additional questions need to be addressed by the authors such as whether the Hho1p/Arp4p-mediated chromatin compaction affects histone modifications, gene transcription or recruitment of ageing-associated factors during RLS, just to cite some.
The reviewer's questions regarding whether Hho1p/Arp4p-mediated chromatin compaction affects histone modifications, gene transcription or recruitment of ageing-associated factors during RLS are very interesting but require extensive transcriptome and proteome analysis. We understand that the interaction between the two chromatin proteins is not convincingly described. We believe and plan further analyses to answer this. Much has been done in the context of the interaction between Arp4p and Hho1p. But here we are the first to show the effects of the arp4 and hho1 mutations on replication ageing, expressed in both as several daughters produced by single mother cells and in the time unit. The genetic interaction seems very much additive and we shall address this in the future.
- In the description of strains genotype, the authors should make clear to the readers how the DY4285 and DY2864 genotypes refer to arp4 and hho1D mutants since both mutations do not appear in the corresponding genotypes. I had to look at the reference to find out that act3 corresponds to arp4 and yll127C to hho, but this information should be provided in this manuscript
We have provided the necessary details in Table 1 and the text. Please see lines:117, 118, 120, Table 1 and 123. We have provided explanations for the genotypes for ease of understanding.
- In comparing the phenotypes between the mutant haploid and heterozygous diploid strains, in most cases the authors claim that since the heterozygotes are wild-type that “one wild type copy…is sufficient to revert to the wild-type phenotype” (i.e Lines 287, 325 ecc…). These conclusions are misleading because they do not imply an underlying genetic interaction (as the reversion/phenotype rescue should do) but only the obvious assumption that the mutations are recessive. The authors should revise it carefully
We are grateful for this suggestion. However, we do not claim that heterozygotes are wild-type. Based on the presented data, it is indisputable that one correct version (wild-type) allele of ARP4 or HHO1 is sufficient to restore the wild-type phenotype. For the identification of these interactions, further molecular analysis is needed, including, for example, transcriptome analyses. An interesting observation is that the ARP4 / arp4 heterozygote also reverts to the wild version (ARP4 is essential). It has previously been demonstrated that ORC1-6 genes involved in replication initiation show significant changes in growth and ageing parameters. Nevertheless, based on the analyses shown in this experimental setup, it appears that genetically ARP4 and HHO1 have recessive gene characteristics since mutation effects are not observed in heterozygotes. To demonstrate the haploid and heterozygous experimental setup, was the overriding purpose of this study.
To answer the reviewer’s suggestion, we have provided the following edits in the text: Please, see lines: 276 “Therefore, we suggested that one wild-type allele of the ARP4 and/or HHO1 genes is sufficient for the cells to revert to the wild-type growth phenotype in a rich fermentative medium (YPD).”
Lines: 328: „This result represented the importance of Hho1p and Arp4p in the maintenance of the proper cell cycle progression during replicative ageing. On the contrary, no major differences were noted between the WT (2n) strain and the three heterozygous chromatin mutant strains, suggesting that one ARP4 and/or HHO1 allele was enough to confer the wild-type phenotype.”
- There are different points in the manuscript where the authors describe that either single or double heterozygotes show growth, replication parameters (i.e line 290) as well polysome profiles (line 466) are different (in some cases even better) than wild-type diploid strains. To the best of my knowledge, these observations are counterintuitive and deserve an accurate description/explanation.
Thank you for these comments. However, the changes in the heterozygote experimental setup suggested by the reviewer are minimal. There is a minimal (statistically insignificant) change in the polysome profile, and there is no change in the cell cycle as well. Furthermore, all ageing parameters are not statistically significant. Also, the mean doubling time does not change as shown in Fig. 2b. The changes that the reviewer observed in Fig. 2a are also minimal, but are not related to the growth dynamics since the mean cell duplication time is identical for heterozygotes and WT (2n). There is no doubt that the double heterozygote and HHO1/HHO1 strains achieve a higher OD, but this is a result of the strain's characteristics. Other strains show similar behaviour, but the mechanism is elusive so far.
We have edited the text in the manuscript to address this suggestion. Please, see lines 281-293 “Previous analyses showed that lack of the HHO1 expression (hho1Δ/hho1Δ strain) leads to disturbed gametogenesis [35]. Interestingly, here, we showed that the presence of only one HHO1 allele leads to a significant escalation in the sporulation frequency in comparison to the wild-type WT (2n) (Fig. 2d). Therefore, it could be speculated that not only the availability but the amount of the Hho1p is decisive for the sporulation efficiency. As previously reported [35], a high binding correlation of Hho1 and Ume6 (the master repressor of early meiotic genes) proteins at promoters of early meiotic genes during vegetative growth was revealed by ChIP-chip. In addition, the authors showed that Hho1 and Ume6 are depleted during meiosis and Hho1p is enriched in mature spores. Authors concluded that “Hho1p may play a dual role during sporulation: Hho1 and Ume6 depletion facilitates the onset of meiosis via activation of Ume6-repressed early meiotic genes, whereas Hho1p enrichment in mature spores contributes to spore genome compaction”. The lesser amount of Hho1p in heterozygotes could facilitate and quicken the activation of early meiotic genes, but this assumption needs further detailed analyses.”
Minor points:
- Images of ChYCA should be provided either in the text or in Supplementary materials
We would have provided comet assay images in the case if the main focus was on the chromatin studies. The chromatin comet assay was not studied in full detail. It was a method for additional investigation of the interaction between the two chromatin proteins. Moreover, we have so many figures and methods that comet images for the eight strains will be overloaded, but if the reviewer insists we would provide them.
Recently, we started to study the role of chromatin in `The RLS of these strains and believe that soon we could provide a more detailed analysis of the role of chromatin in this process, especially in the context of genetic mutations.
References 20 and 26 are identical.
Thank you for this suggestion. The references are arranged.

Reviewer 2 Report
Physical interaction between the actin related protein (Arp4p) and the linker histone H1 (Hho1p) is important in yeast, it maintains chromatin structure, genome stability and cellular morphology. In this article, the authors examine the role of this interaction in chronological ageing and Replicative life span.
They used haploid and diploid yeast to have an homozygous and a heterozygous condition. They conclude that genetic interactions of arp4 and Hho1p alleles affect lifespan and ageing.
The article is well written, the experiments are well presented but some of the conclusions are not definitely supported by the data. Prior to publication the authros need to address the following:
Comments:
In figure 1a the growth rate of the wild type vs the haploid mutant strain is shown. The text indicates that the single arp4 mutant had the same growth rate as the wild type, next on Figure 1b, the authors measure the number of cells per ml in the culture media and the arp4 haploid mutant has less cells per ml than the wt. The authors say this indicates that the haploid mutant cells take a longer time to grow.
Please clarify:
1.- Why are there less cells per ml of culture in Fig 1b in the arp4 haploid mutant but a higher OD in Figure 1a? Aren´t ODs a direct measure of the number of cells in the media?
2.- The time between 4 and 6 hours of cell culture seems also to be determinant, can you comment on what happens at that period of time that seems important in this assay?.
3.- It is clear that the strain that carries both mutant genes (arp4,hho1) presents a different growth rate than the haploid or the wild type strain, but in figure 1c it looks like there is no difference between the arp4 haploid and the double mutant. Can you further comment on these effect? Does this mean that the hho1 histone is not necessary for cell growth?
In figure 2. The authors analyzed the role of the heterozygous mutants in growth rate but also in gametogenesis. They found that Hho1 mutants affected sporulation efficiency.
Can the authors comment whether this is a normal response to replicative stress? Also, they put three asterisks in the last bar of figure 2d, however, the effect of the double heterozygous seems mild, since the error bars overlap with the heterozygous Hho mutant. Can you provide a P value for this experiment?
Also, have you looked at the IME1 gene expression? If you remove Hho1 is it possible that this gene might be upregulated, could you comment on that at the discussion section?
For figure 3 please clarify:
The mutants seem to stall at G1 phase, can you confirm these are the same cells that proceed to the G2-M phase? These mutant cells do not die?. I ask this because of the results of Figure 2d in which you have more sporulation in the Hho1 mutants.
In line 338: Please rephrase the sentence. “To see how the global chromatin compaction changes its compaction” you can try: To evaluate chromatin compaction changes in….
In line 349: Please rephrase the sentence: “…and the diploid yeast strains were observed during the observed period….”
In figure 4, the authors say that chromatin compaction was affected, however it is not clear to me what statistical analysis they used. The authors need to perform a statistical analysis to demonstrate the differences are significant (Mann-Whitney maybe?). Also, this global effect on chromatin might be observed better in a non-rich media.
In figures 5 and 6, the authors’ evaluate the life-span of the mutants (haploid or diploid), The phenotypes that differ from the wild type, all seem to derivate from the arp4 mutant. It would be interesting to see if the observed mutant phenotypes, are rescued only by the hho gene. Is it possible to have a diploid that only has a wild-type copy of the hho1 gene but is null for arp4?.
It looks like these phenotypes depend only on arp4 and that there is little or non-participation of hho1.
Therefore the statement .“This allowed us to conclude that Arp4p and its interaction with the yeast linker histone Hho1p are key players that regulate not only DNA stability but also ageing” should be changed because the data indicate that these phenotypes depend on an arp4 independent function and not to its association with hho1.
On figure 8 please show the vacuole with an arrow and the “wrinkled” phenotype with an arrow head or asterisk.
For figures 9 and 10, in the text. Line 441 please show which one is the treatment with zeocin and how did you evaluated DNA repair.
I do not agree with the statement at line 444-445 in which it says that the haploid double mutant was the most sensitive strain in all the treatments. Looking at all the data, the double mutant phenocopies the arp4 haploid mutant but not the hho1 haploid mutant, even in some treatments, the hho1 mutant is unaffected, therefore we cannot assume that the phenotypes observed are all derived from a lack of interaction between these two proteins.
In Figure 12b. The data can also be interpreted as a lack of response to the treatment which is strange because the haploids respond like the wildtype (with different levels of course), but in the diploids, the response is affected. Please can you comment on this effect in the discussion section?
I do not completely agree with the statement at lines 615-617. The data shows that it is mostly dependent on arp4. Maybe there is some involvement of Hho1 shown in the genetic interactions, but is seems to be mild. Perhaps if you use another allele (one that specifically targets the interaction sites in the protein) you can actually see without any doubt that the phenotypes are indeed due to the interaction of arp4 and Hho1 alleles.
In the discussion section you also mention that there are many copies of Hho1 gene. Do you think that there is a compensation from these other copies?
Author Response
Review form 2
Thank you very much for the detailed review of our work!
It indeed helped us a lot in improving it. Moreover, it further granted us the possibility to improve our work.
We tried to address all questions and suggestions. Please, see below:
- In figure 1a the growth rate of the wild type vs the haploid mutant strain is shown. The text indicates that the single arp4 mutant had the same growth rate as the wild type, next on Figure 1b, the authors measure the number of cells per ml in the culture media and the arp4 haploid mutant has less cells per ml than the wt. The authors say this indicates that the haploid mutant cells take a longer time to grow.
Please clarify:
1.- Why are there less cells per ml of culture in Fig 1b in the arp4 haploid mutant but a higher OD in Figure 1a? Aren´t ODs a direct measure of the number of cells in the media?
Thank you for these suggestions and questions. Yes, we have shown that two different approaches to measuring growth rate show a different result. We observed that arp4 grew slower in liquid cultures, but the growth curve (OD measurement) did not show this. This is because OD measures the number of all cells (viable and dead), and also depends on the size of cells in the culture. Our previous data showed differences in the size of arp4 and wild type cells. Earlier it was shown that OD is not always a good measure to show the yeast rate of growth.
2.- The time between 4 and 6 hours of cell culture seems also to be determinant, can you comment on what happens at that period of time that seems important in this assay?
Indeed, between the monitored 4th and 6th hours, we observed an acceleration in the growth rate, but it is difficult to say unequivocally what is the determinant of it. It is possible that in the future such analyzes should be performed more frequently, e.g. every hour. In the context of cell doubling time analyses, this is rather irrelevant. Fig. 1c is significant, which unambiguously confirms that both arp4 and the double mutant have a significantly extended doubling time.
3.- It is clear that the strain that carries both mutant genes (arp4,hho1) presents a different growth rate than the haploid or the wild type strain, but in figure 1c it looks like there is no difference between the arp4 haploid and the double mutant. Can you further comment on these effect? Does this mean that the hho1 histone is not necessary for cell growth?
Yes, the same is also seen in the ageing analyzes. Arp4 appears to be crucial in determining many aspects of yeast biology in the context of the interaction of Hho1 and Arp4 proteins. Our previous data indicate that when cells grow in optimal conditions the knockout of the HHO1 gene has almost no effect, however, the growth of hho1delta cells in minimal medium or at stress conditions is affected in comparison to that of the wild type
- In figure 2. The authors analyzed the role of the heterozygous mutants in growth rate but also in gametogenesis. They found that Hho1 mutants affected sporulation efficiency.
Can the authors comment whether this is a normal response to replicative stress? Also, they put three asterisks in the last bar of figure 2d, however, the effect of the double heterozygous seems mild, since the error bars overlap with the heterozygous Hho mutant. Can you provide a P value for this experiment?
Thank you for this suggestion. This is our oversight. Indeed, here both HHO1/hho1 and the 2n double mutant have increased sporulation efficiency which is statistically significant for p < 0.001 compared to WT. That's why we changed the text in Lines 294:
“Our results further prove that the combination of the two mutations (arp4 and hho1Δ) in one genome (ARP4 HHO1/arp4 hho1Δ cells) or HHO1/hho1Δ heterozygote, regardless of the presence of a wild type allele, did significantly affect the sporulation efficiency (p < 0.001). The heterozygous diploid double mutant ARP4 HHO1/arp4 hho1Δ cells produced three times more and slightly fewer tetrads compared to the ARP4/arp4 and HHO1/hho1Δ heterozygotes, respectively.”
Sporulation is a key process necessary for yeast to survive in natural conditions. Further, data showed, that in SWI6/swi6Δ cells, sporulation decreased significantly (15-fold) compared to that of the WT diploid, whereas the swi6Δ/swi6Δ strain did not sporulate at all (doi.org/10.1242/jcs.226480). Our recent studies have shown that in the case of ORC1-6 (origin replication complex) heterozygotes there is a slight reduction in sporulation frequency (10.3390/cells11081252). Therefore, the result presented in this paper is interesting and provides new insight into sporulation. It is interesting to note that sporulation frequency increases in the heterozygote setup HHO1/HHO1. Our goal is to demonstrate this effect in the future at the level of hho1/hho1 homozygote as well.
Also, have you looked at the IME1 gene expression? If you remove Hho1 is it possible that this gene might be upregulated, could you comment on that at the discussion section?
We appreciate this suggestion. However, we did not determine the expression of IME1, so it is difficult for us to discuss this data. The gametogenesis result complements our general knowledge of the histone linker mutation. Nevertheless, thank you for this suggestion, we will use it in the next paper.
- For figure 3 please clarify:
The mutants seem to stall at G1 phase, can you confirm these are the same cells that proceed to the G2-M phase? These mutant cells do not die? I ask this because of the results of Figure 2d in which you have more sporulation in the Hho1 mutants.
Yes, in cultures of haploid strains there is an enrichment of G0-G1 cells, but these cells can not sporulate. Only diploids can sporulate and the HHO1/hho1Δ heterozygotes do demonstrate higher sporulation efficiency with an explanation given in the previous paragraph.
Indeed, G0/G1 accumulation of cells is important for the longevity of cells though. According to previous reports, after the diauxic shift (around the 14th-24th hour of cultivation), two distinguishable cell populations are presented in the G1 phase of yeast S. cerevisiae cultures, quiescent (Q) population in which daughter cells are predominant and are the longest-lived and non Q population in which mother cells are predominant and are the shorter-lived (Allen et al., 2006; Li et al., 2013). On the fourth hour for all haploid cultures studied, the largest number of cells was in phase G1 as after inoculation of overnight culture into fresh medium, they are still in lag phase, preparing for cell division. Importantly, the kinetics of the G0/G1 to S/G2 transition is different for the studied haploid strains. The higher percentage of Q cells the faster transition from G0/G1 to G2 would be. The most distinctive delay was observed for the arp4hho1delta strain. We could speculate that although stationary cultures of these mutants have а high percentage of G1 phase cells, they are rather nonQ, non-long-lived cells, and therefore poor survivors. This in turn raises the question of the relationship between the ability of chromatin mutants to form a proper quiescent population and the replicative lifespan and the need for further research to elucidate it. both, the cellular viability and the percentage of G2/M cell fraction. We find this intriguing and deserving our attention results. We hope that the reviewer will accept our logic and will grant us the possibility to publish and disseminate these results in the field.
In line 338: Please rephrase the sentence. “To see how the global chromatin compaction changes its compaction” you can try: To evaluate chromatin compaction changes in….
We have edited the text: Lines 342: “To assess changes in global chromatin compaction in the haploid and diploid sets of the four strains studied, during their replicative ageing, we performed Chromatin Yeast Comet Assay (ChYCA).”
Thank you!
In line 349: Please rephrase the sentence: “…and the diploid yeast strains were observed during the observed period….”
Thank you for this suggestion. In Line 351 it is already edited accordingly: “Subtle changes in global chromatin organization between the haploid and the diploid yeast strains were observed during the examined period of their RLS.”
- In figure 4, the authors say that chromatin compaction was affected, however it is not clear to me what statistical analysis they used. The authors need to perform a statistical analysis to demonstrate the differences are significant (Mann-Whitney maybe?). Also, this global effect on chromatin might be observed better in a non-rich media.
Indeed, here we report the effect on global chromatin compaction of the eight studied strains in YPD cultures to investigate the replicative ageing cells that must be grown in rich media. The chromatin compaction of the four haploid strains in a non-rich media (SD, synthetic defined medium) in the course of chronological lifespan was reported previously (Vasileva et al, 2021). A similar tendency was observed at the 4th hour.
Statistical significance is provided.
Thank you!
- In figures 5 and 6, the authors’ evaluate the life-span of the mutants (haploid or diploid), The phenotypes that differ from the wild type, all seem to derivate from the arp4 mutant. It would be interesting to see if the observed mutant phenotypes, are rescued only by the hho gene. Is it possible to have a diploid that only has a wild-type copy of the hho1 gene but is null for arp4?.
That's an interesting question. Only haploids showed differences in the reproductive capacity while in diploids the presence of a wild-type allele of ARP4 and/or HHO1 cover this phenotype. ARP4 is an essential gene and a null mutant is not viable.
It looks like these phenotypes depend only on arp4 and that there is little or non-participation of hho1.
Therefore the statement: “This allowed us to conclude that Arp4p and its interaction with the yeast linker histone Hho1p are key players that regulate not only DNA stability but also ageing” should be changed because the data indicate that these phenotypes depend on an arp4 independent function and not to its association with hho1.
We hypothesize that if these phenotypes depend on an arp4 independent function and not on its association with hho1 one could expect to overlap the curves of arp4 and arp4hho1Δ mutants. The curves of the two strains do not overlap and cross. The combination of the two mutations, arp4 and hho1Δ, in one genome results in a unique, characteristic curve for the double mutant. This suggests some genetic interaction. Maybe arp4 has a stronger influence on the phenotype of the double mutant, but the HHO1 deletion exerts an epistatic effect on the arp4 phenotype. Our previous data also demonstrated a complicated interaction between the two genes with an epistatic effect for some phenotypes and synergistic for others. Thus, the arp4 and arp4hho1Δ phenotypes are not identical and the differences should be due to the knockout of the HHO1 gene.
Until we are sure of all these interactions, we changed the statement in Line 402 to “This allowed us to conclude that Arp4p is the key player that regulates not only DNA stability but also ageing.”
- In figure 8 please show the vacuole with an arrow and the “wrinkled” phenotype with an arrowhead or asterisk.
These are not wrinkled vacuoles, but rather large vacuoles (this is our oversight, so thank you for your attention). That's why we changed that too. We have shown large single vacuoles in Fig. 8 as was suggested by the reviewer.
- For figures 9 and 10, in the text. Line 441 please show which one is the treatment with zeocin and how did you evaluated DNA repair.
We changed this statement Lines 441: “The toxicological analyses also showed that mutations in the studied genes did not affect the effectiveness of DNA repair after treatment with zeocin (data not shown).”
In Lines 440 we provided the following: “In the heterozygous set, only ARP4/arp4 differed in growth on the G418 while the double mutant showed resistance to G418 comparable to that of the wild type. This again demonstrates the effect of the simultaneous presence of both mutations in one genome and the influence of hho1 deletion on the double mutant phenotype (Fig.10).”
We did not evaluate DNA repair here, but we assumed that cell growth on the zeocin did not lead to DNA damage. Zeocin causes cell death by intercalating into DNA and inducing double-stranded breaks of the DNA.
- I do not agree with the statement at line 444-445 in which it says that the haploid double mutant was the most sensitive strain in all the treatments. Looking at all the data, the double mutant phenocopies the arp4 haploid mutant but not the hho1 haploid mutant, even in some treatments, the hho1 mutant is unaffected, therefore we cannot assume that the phenotypes observed are all derived from a lack of interaction between these two proteins.
We generally agree with this reviewer's statement. Nevertheless, the double mutant is more sensitive to stressors than arp4. To mitigate this statement, we have changed this sentence.
As previously reported and commented, in general, the phenotype of HHO1 deletion is hardly detectable on rich media. Overall, as expected the haploid strains were more sensitive to the applied treatments than the heterodiploids, with the haploid arp4 and double mutant arp4 hho1Δ being the most sensitive strains.
In Lines 447 we provided the following text: “Overall, as expected the haploid strains were more sensitive to the applied treatments than the heterodiploid counterparts (Fig. 9 vs Fig. 10). The haploid double mutant arp4 hho1Δ displayed higher sensitivity to Congo red, CFW (25 μg/ml), NaCl and Camptothecin than the arp4 that reflects the influence of HHO1 deletion. Also, the arp4 hho1Δ was more sensitive to G418 (50 μg/ml) in comparison to the WT and hho1Δ strains but more resistant compared to the arp4 mutant (Fig. 9) an indication for the Hho1 depletion is suppressive to Arp4 mutation.”
- In Figure 12b. The data can also be interpreted as a lack of response to the treatment which is strange because the haploids respond like the wildtype (with different levels of course), but in the diploids, the response is affected. Please can you comment on this effect in the discussion section?
We've reinforced our discussion of the excerpt. Please see Lines: 623-628: “Nevertheless, heterozygotes displayed lower levels of ROS than wild-type strains. Recent studies evaluated oxidative stress adaptation mechanisms after deleting genes required for growth. It has been found that gene loss can enhance an organism's ability to evolve and adapt. According to these findings, the loss of a single gene can facilitate adaptation through the opening of alternative evolutionary pathways (10.1093/molbev/msaa172).”
- I do not completely agree with the statement in lines 615-617. The data shows that it is mostly dependent on arp4. Maybe there is some involvement of Hho1 shown in the genetic interactions but it seems to be mild. Perhaps if you use another allele (one that specifically targets the interaction sites in the protein) you can see without any doubt that the phenotypes are indeed due to the interaction of arp4 and Hho1 alleles.
The deletion of only the HHO1 gene has a negligible effect if any on the phenotypes of mutant cells in rich media and under optimal conditions. However, in combination with a mutation in the ARP4 gene the resultant double mutant does not copy the phenotype of arp4 that is expected if HHO1 deletion has no consequence. In most cases, the phenotypes of the double mutant strain were well distinguishable from those of the single arp4 mutant sometimes being in the middle between the arp4 and hho1 mostly resembling the phenotype of one of the single mutants all that indicating the influence of hho1 deletion to modulate the effect of arp4 mutation.
In the discussion section you also mention that there are many copies of Hho1 gene. Do you think that there is a compensation from these other copies?
We have corrected our stylistic errors and replaced copies of the genes with alleles as it should be. And mRNA transcripts, not copies.
Please see lines: 581, 582 and 599.
Thank you again for the thorough review! It certainly helped us perceive the errors and unanswered questions in our work.

Round 2
Reviewer 1 Report
I am still convinced that the fact that heterozygotes show a similar growth kinetics compared to homozygotes indicates that the mutations are recessive. Anyway, I suggest that the on line 304, the sentence " ...for the cells to revert to wild-type growth phenotype..." should be replaced by "...for the cells to confer a wild-type growth phenotype..."
Author Response
Dear reviewer,
Thank you very much for the positive evaluation of our work and for all the comments you have provided.
They indeed increased the quality of our work and paved the road to the project's future development.
The comment:
"I am still convinced that the fact that heterozygotes show similar growth kinetics compared to homozygotes indicates that the mutations are recessive. Anyway, I suggest that on line 304, the sentence " ...for the cells to revert to wild-type growth phenotype..." should be replaced by "...for the cells to confer a wild-type growth phenotype...",
It is taken into account. In Line 277 the sentence is edited accordingly and is marked by track changes. The number of the Line is different from 304 as the format of the manuscript changed the numbering.
Thank you again,
Warm regards,
Milena Georgieva
Mateusz Molon
Reviewer 2 Report
I had already accepted the manuscript without any modification.
Author Response
Dear reviewer,
Thank you very much indeed for the positive evaluation of our work.
The comments and all provided suggestions improved the quality of our manuscript a lot.
Warm regards,
Milena Georgieva
Mateusz Molon